# Characterization and Efficient Management of Big Data in IoT-Driven Smart City Development

**DOI:** 10.3390/s19112430

**Published:** 2019-05-28

**Authors:** Alaa Alsaig, Vangalur Alagar, Zaki Chammaa, Nematollaah Shiri

**Affiliations:** 1Concordia University, 1455 De Maisonneuve Boul. W, Montreal, QC H3G 1M8, Canada; alaasaig@hotmail.com (A.A.); zakichammaa@gmail.com (Z.C.); shiri@cse.concordia.ca (N.S.); 2Jeddart University, Hamzah Ibn Al Qasim Street, Al Sharafeyah, Jeddah 23218, Saudi Arabia

**Keywords:** smart cities, big data, data model, databases, data integration, internet of things (IoT)

## Abstract

Smart city is an emerging initiative for integrating Information and Communication Technologies (ICT) in effective ways to support development of smart cities with enhanced quality of life for its citizens through safe and secure context-aware services. Major technical challenges to realize smart cities include resource use optimization, service delivery without interruption at all times in all aspects, minimization of costs, and reduction of resource consumption. To address these challenges, new techniques and technologies are required for modeling and processing the big data generated and used through the underlying Internet of Things (IoT). To this end, we propose a data-centric approach to IoT in conceptualizing the “things” from a service-oriented perspective and investigate efficient ways to identify, integrate, and manage big data. The data-centric approach is expected to better support efficient management of data with complexities inherent in IoT-generated big data. Furthermore, it supports efficient and scalable query processing and reasoning techniques required in development of smart city applications. This article redresses the literature and contributes to the foundations of smart cities applications.

## 1. Introduction

Rapid urbanization all over the world over the last 20 years has increased the level of demand on sustainable economic growth and high quality of life in urban areas [1,2]. To meet these demands, several critical infrastructure components, such as city administration, education, health care, public safety, utilities, and transportation must be networked and managed more “intelligently”. The notion of “Smart City” emerged from urbanization. Its mission includes optimization of resources, minimization of wastage and pollution, and improving the quality of life of its citizens using physical and cyber elements in ICT. To realize “smartness” [3], advanced techniques based on ICT must be explored to sense, gather, and integrate the key information of core systems of criticality for the city, i.e., ICT should innovate to build “smart computing” based on the triangular web *Instrumentation, Interconnectedness, Intelligence* ($I_{3})$. The primary goal of this paper is to investigate the nature of “data” that is common to $I_{3}$, and explore the appropriate database support for storing, managing, integrating, and analyzing data for critical applications of a smart city.

Instrumentation in $I_{3}$ refers to an integrated sensor system, in which sensors, wearables, and other interoperable devices are connected in a large network. Such a network includes internal network of sensors, external connections through wireless networks, and perhaps special sensors to fetch data from numerous remote data sources. The network architecture may include specialized sub-nets, where each sub-net is specialized in networking instrumentations for use in one application domain. This conceptualization has lead researchers [4,5,6] to consider Internet of Things (IoT) [7,8] as a basic framework to conceive smart cities. For achieving correctness and efficiency, it is necessary to capture the knowledge about the “things” in this IoT and specify their services in efficient data stores so that it can be shared by the three actors [9] *Research and Development* (ICT), *Service Industry* (provider/consumer), and *Government* (policy developers) who will contribute to smart city development. Because the expertise and expectations of these actors are different, the data stores must provide intelligent interfaces for the actors to browse, select, and use the stored knowledge. There is another dimension to the information gathered by the “things”, namely “meta information” that qualifies data/information/knowledge. For example, the humidity level sensed by Humidity monitor is data, whereas the aggregation of time and location information sensed by the sensor in that device constitutes the context, which qualifies the observed humidity information. Thus, in general observations may enable the system to construct *contexts* (meta information) in which certain observations or computations or communications or adaptations happen in the system.

It is essential to understand that “data” is to be used for computing, and context is to be used for adding *context-awareness* to computations, i.e., the system is made “smart” by being aware of itself and its environment in order that it can adapt itself to changing environment. Consequently, the data store must be organized to disambiguate between data and context. Interconnectedness brings in not only network topology but also protocols and access controls for safe and secure communication. This knowledge must be part of data stores. While context-awareness is part of intelligence, smartness has to mean “being intelligent and be able to infer to adapt and react to dynamically changing situations”. Hence, database should include a knowledge base module in which facts (observed) and rules for inferencing consequences, and adaptation policies are stored separately from “data” used for general computations. Thus, we are led to consider an intelligent data base support to drive data-centric and context-aware computations in IoT-driven smart city development. The Big Data (BD) characterization of IoT-driven data and abstract modeling of “things” in IoT that we discuss in this paper form a firm basis to achieving the primary goal stated earlier. Although some recent works [4,5] have mentioned the important characteristics of IoT data and call for using data engineering methods to optimize existing infrastructure in order to improve services in smart cities, they have given neither any data-centric methodology nor specific data store architecture for achieving the goal. The contribution that we make in this paper, itemized below, answers this open problem.
*IoT–BD Characterization:* In Section 2 we characterize the BD driven by the IoT of a Smart City using the V model. The significance is that no such comprehensive investigations on the characteristics of BD that impact on the choice of data store for smart city development has ever been reported.*Database Choice:* In Section 3 we critically study the features of relational (SQL) and NoSQL databases schemas from the perspective of supporting BD, as characterized in Section 2, and justify MongoDB [10] as a suitable database to manage the IoT-driven BD. We should remark that our investigation, based on the V-model satisfaction, has not been reported in related literature.*Smart City Development:* In Section 4.1 we compare our work with other published works. In Section 4.2 we discuss an abstract architecture for smart city development, currently under implementation [11], and explain the interactions between the data stores and other architectural elements in fulfilling their expected goals.

## 2. IoT-driven BD

The goal of this section is to motivate the importance of a suitable database that can adequately handle the complexities of the V model of BD in smart city development. Towards this, we briefly review IoT basics, and characterize the “things” of Smart City IoT using the V model of BD.

The term IoT was coined by the authors of the Telecommunication Union report [8] and MIT [7] to refer to such a network as IoT. In 2009 Ashton [12] predicted that IoT has “the potential to change the world, just as the Internet did. May be even more so”. Since then, IoT has been rapidly expanding to a vast ecosystem of technologies, machines, humans, and computing-communication-control devices. Although IoT definitions and requirements are numerous and no one is complete, we can embrace the most comprehensive descriptions variety, accuracy, timeliness, context, reliability, and trust [13,14]. In particular, smart cities need to have their sensors, used by different application domains (industrial, social, economic, …), to be connected either in a central unit or distributed network to manage contextual data [6]. From the perspective of modeling IoT we draw upon the two basic concepts, namely *part-of* and *Is-a* from Object Oriented Design (OOD) and Entity Relationship model (ER) of database design. “Things” are *part of* IoT and each physical device such as sensor, actuator, or RFID *Is-a* “thing”.

Among the “things” of IoT, the most fundamental “things” of importance for Smart City are sensors and actuators. A sensor is a device that converts physical parameters into electronic signals, which can be interpreted by humans or can be stored in autonomous system [6]. Actuators provide a mechanical response according to the input provided by the sensors and are usually processed by other electronics or mechanical devices. In some cases, a sensor can be used as an actuator and vice-versa; as is the case with a few microphones and motors. In IoT sensor networks provide an entry to the system, and actuator configurations enable the system to make changes to its environment. Sensors can be broken down into three main categories, as suggested in [6].
**Technical Sensors:** The three types are the following:-**Environmental Sensors:** These are used in Environmental Monitoring or Urban Sensing. Some applications are in Meteorology and Weather monitoring, Air Pollution Quality Monitoring, Heat Island Detection, Flood Monitoring, and Nuclear Radiation Safety.-**Mobile Sensors:** Wearable ambient Sensors, Mobile, and Sensor Web are different terms to describe the same category. They are used in Ubiquitous Measurement, and Disaster Management.-**Pervasive Sensors:** They are used for ubiquitous sensing, and socially aware computing. Applications using this type of sensors are Smart context-aware Environment and Home, Ambient/Active Assisted Living, Pervasive Healthcare, RFID-Based Location and Tracking, and Socially Aware Computing.**Technical Sensors-Remote Sensors:** This type of sensors is mainly for remote sensing, from satellite-based to terrestrial. Applications domains of this type of sensors include “Classic” Airborne and Space-borne Optical System, and Atmosphere/Aerosols.**Human Sensors:** These are of two kinds.-**People as Sensors:** Examples of this type of sensors include Citizens as sensors, human sensing, physiological sensors, wearable body sensors, participatory sensors, and Volunteered Geographic Information (VGI). They are primarily used for Flood Monitoring, as Sensing Platform in Smart Cities, and for Physiological Parameters such as saturation, stress levels, and Noise Mapping.-**Collective Sensors:** These include Mobile phone sensing, crowd sensing, social sensing, online sensing, and social media. These types are used for Disaster and Incident Management, Mobility Patterns and Transportation, Socio-Physical Context Estimation, Tourism, Epidemiology, and Disease Detection.

Similar classification could be given for other “things” in the IoT; however, this classification is based on the V model of BD.

### 2.1. A Classification of “Things” Using the V model of BD

The integration of Smart City applications has to be aware of the characteristics of the “things” involved in each application. Among the various Vs suggested in the literature that characterize BD, the following 10 have been selected as the most comprehensive [15]. We explain how the “things” inherit these Vs in a natural manner.
**Volume:** We refer to the size of IoT as its volume. IoT comprises of many “things”, some of which may be interconnected, yet may act autonomously. The “interconnectedness” reflects the Vincularity (relationship) among the “things”, i.e., if “things” X and Y are connected, it is obvious that the relationship requires is inherent in it. Not only the number of “things” in IoT will be large, but also the number of the types and size of such relationships.**Vincularity:** This refers to the challenge related to the process of connecting two or more different “things” in order to achieve a complex service. The Vincularity process requires the accuracy of “things” in the fusion.**Variety:** It refers to the heterogeneity of “things” in IoT. Different types of “things” such as sensors, actuators, medical devices, and human actors generate and consume different types of information generated through “things”. They also have different sets of capabilities for storing, processing, and reasoning with data.**Velocity:** It refers to the “streaming” of data at different rates and the ability of sensors/actuators to process them. Periodic data flow, asynchronous functioning ability of “things” in the network, and the unpredictable volume of data flow at different periods add complexity to store and manage data. Because IoT assets will in general be distributed, and processing video and digital images require contextual information [16], the velocity characteristic raises the complexity of real-time streaming of BD.**Veracity:** It refers to the correctness and relevance of data generated by a thing (device or sensor). Some data may have “time constraints” that restricts the utility of data. A “thing” in IoT may be either compromised or damaged. In either case the data is deemed incorrect.**Validity:** It refers to the “relevance” of a “thing” with respect to the achievement of goals. A device may work correctly and emit correct data (veracity is satisfied); however, that may not be necessary to achieve the stated goals. In that case, it is not valid. Please note that a thing that is valid satisfies veracity criterion, but the converse is not true.**Vitality:** It refers to the “criticality” of the thing in IoT. A thing is either being a sensor, an actuator, or a medical device, can be labelled as “mission critical” if the information processed by it is critical. Vitality characteristic implies both veracity and validity; however, the converse is not true.**Value:** Everything in IoT must have “high value”, in the sense that it must contribute to the overall purpose of the IoT. A thing that does not contribute to the goal of IoT has no value and must be ignored or removed. Value of a thing is enhanced only if it is authentic and dependable, where “authenticity” must be certified by an Intelligent Trusted Authority (ITA) of the IoT. Dependability as defined in [14] is “the ability to provide services that can be justifiably be trusted by the client “things” in the IoT”.**Volatility:** Every piece of data must have a specific “lifetime” defined by the system developers depending on where and how it will be used. Volatility characteristic motivates the avoidance of storing meaningless data.**Visualization:** It refers to the management of data presentation. When data is required to be fused or exchanged between two things, where the views on data of the things are not the same a presentation conversion is required. It is important to define the methodology or the proper presentation of the data to the thing requiring the data. The existence of other characteristics will increase the difficulty of meaningful presentation.

From the above classification we can observe the following characteristics with respect to data generated and consumed by the “things”.
*Data Generation:* Volume, velocity, variety, vitality, and vincularity are essential V-attributes of data generated by the “things”.*Data Quality:* Veracity, validity, value, and volatility are essential V-attributes for quality of data consumed by the “things”.*Data Transfer:* Vincularity, velocity, validity, volatility, and visualization are essential V-attributes for data exchange/sharing among the “things”.

### 2.2. A Layer-Centric Dominance of the Vs

In the literature [17,18], a layered IoT architecture is studied as the basis for implementing an IoT application. The five layers are *Device Layer* (DL: combines things and Edge technology), *Access Gateway Layer* (AL: data preprocessing, protocol conversion), *Internet Layer* (IL: cyber communication), *Middleware Layer* (ML), and *Application Layer* (AL). The layer DL is the Edge that consists of “things” that collects information and perform a limited amount of preprocessing of data. The layer AL is the Fog that handles subscribe/publish and message routing, and enables communication across different computing platforms. The IL enables communication across several sub-nets and distribution of data and messages to remote sites. The ML performs data storage, data analysis, and decision-making. The layer AL is responsible for service discovery, service composition, and service delivery. In several applications, Cloud is used to combine the roles of ML and AL layers. These 5 layers can be mapped into the layers of a 4-tier architecture [15] that is often used for BD applications. This mapping, combined with the mapping of the Vs to the 4-tired architecture, convinces us the importance of data store selection for IoT-driven BD for Smart City development.

***BD Layer-1:*** This is the Storage Layer where general as well as application-specific data is stored by developers. The 5 most dominant V’s that have an impact on storage are *Value*, *Volatility, Variety, Volume*, and *Velocity*. It is a challenge, when faced with a huge amount of data, to specify a value for every piece of data. Without due respect to the value feature, huge amount of storage could be allocated to data whose value is low (or unknown). It is essential to understand how Volatility affects and is affected by Volume, Variety, and Velocity. The three distinct ways in which Variety impacts storage are (1) types of data, such as text, image, and graphs, (2) functionalities for manipulating data, and (3) heterogeneous data formats. The study [19] reports that about 5% of collected data represent the structured data, and the rest 95% is semi-free structured. The Volume feature is a primary concern for devising storage structures. Questions such as (1) which data should be centralized, (2) what data should be distributed and shared through access controls, and (3) what storage schemas promote efficient query processing are all primarily affected by Volume characteristic. Velocity creates two challenges at the storage level. One is the necessity to “acknowledge” the receipt of data chunks, which needs to be done periodically and in a timely manner. The other challenge is the necessity to apply some analysis to decide whether to accept the data, and where it should be stored, and for how long it should be stored. It is easy to observe that all 5 IoT layers need storage structures, some centralized and some distributed. Hence, the 5 Vs stated above affect all layers of IoT architecture.

***BD Layer-2:*** In this “Logic Layer”, data processing, analysis, and decision-making happen. Hence, the characteristics *Veracity, Vincularity*, and *Volatility* affect IoT quality. It is essential to guarantee that the source “thing” provides accurate data every time it is queried. Otherwise, uncertain, incomplete, or imprecise data from one source “thing” could cause many errors during data integration. Hence, Veracity of data sources is a dominant characteristic for analysis and decision-making process. With huge data streaming from different sources it is challenging to know the correlation between sources and build associations among data. This shows the importance of Vincularity characteristic in this layer. For the sake of efficient processing, namely improving the response time of answering subsequent requests, it is necessary to save the results of a frequently occurring query in the data stores. If the Volatility of data source is high, it is necessary to evaluate queries every time it occurs in the analysis. The three layers DL, AL, and ML can be regarded as “logical layer” for BD processing. Hence, the 3 Vs stated above affect these three IoT layers.

***BD Layer-3:*** In this “Application Layer” the dominant characteristics are *Validity* and *Vitality* because they are related to data quality and accuracy. A data might be accurate, yet it may not be valid for an application. Hence, after integrating correct data for a specific service, a validity check must be done to ensure that the result meets the expectation of service requester. Validity checking might include semantic checking, and applying policies for validating the authenticity of data delivery to users. Vitality is a dominant characteristic for “critical application domains”, in which validation processes are inadequate to assure accuracy and correctness of data. Depending upon the level of criticality, we may require safety, security, privacy, and ethical analysis of resulting data. This layer is identical to the IoT layer AL.

***BD Layer-4:*** In this “Presentation Layer” data visualization takes place and hence *Visualization* is the dominant feature. For human consumption of data *Visualization* is the only relevant feature. However, if the data is sent to *actuators* which will execute changes in the environment the additional features *Veracity*, *Velocity, Validity*, and *Vitality* also become relevant. As an example, in e-Health the system might send data to physicians as well as to other mechanical devices that can perform a statistical analysis on the data. In such instances, all the five features *Visualization*, *Veracity, Velocity, Validity*, and *Vitality* are dominant. If the IoT layer AL may include humans, we conclude that both IoT layers DL and AL are affected by the above 5 Vs.

From the above discussion that characterizes the Vs of BD and their pervasiveness in IoT implementation layers, it is evident that in order to implement the 5-layered IoT, we need data stores that can effectively handle the V-characteristics of BD.

## 3. Database Support for Smart City Development

The two primary requirements in development of Smart City applications are (1) storing and managing BD, and (2) integrating the data for analysis and decision-making. We need to store the structure of“things”, the services they provide, and their dependencies. We discuss these aspects in the first three sections below. Following that, we evaluate the suitability of existing database models for storing and managing BD and choose one NoSQL model that best suits all the V-characteristics of BD. Finally, we illustrate the adequacy of this database for data integration purposes in a smart city development.

### 3.1. “Things” Model

Our first contribution is a generic and flexible model of “things”. This model is based on *Service-oriented Paradigm* (SoP) [10,20], in order that the IoT layer AL can effectively use the database of “things” for service composition. We view a “thing” in terms of “what it is”, “what services it provides”, and “what are the constraints for providing its services”. In SoP, each “thing” consists of *service functionality* and *service contract*. The former includes a precise description of service and all data that are necessary to generate and provide the service. The latter includes a description of Quality of Service (QoS) attributes to be regarded as a contract between the service provider (the supplier of thing) and service requester (Smart City developer). These attributes include *Context* information, *Legal-Rules*, and *Trustworthiness properties*. Based on SoP, we propose the “thing” schema shown in Figure 1. Its implementation model is shown in Figure 2. The significant features of this schema include *generality*, *extendability*, and *flexibility*. It is general because the schema can describe “things” of many different types. It is extendable because more service and/or contract attributes can be added independent of each other. It is flexible because by fixing the service functionality, the contract part can be changed in the schema, thus providing numerous models of “things” which provide the same service but have different QoS properties. Below we describe service and contract parts of a “thing” in greater detail.

#### Modeling Service Part of “Things”

The elements that describe the service of a “thing” can be classified as *Service Functionality* (SF), *Service Attributes* (SA), and *Service Non-Functionality* (NF). SF is specified by its *Pre-condition, Post-condition, Signature, and Result*. The *Pre-condition* is the condition that must be met by Service Provider (SP) (may be an application client) using this particular “thing” to implement the service. The *Post-condition* specifies what SP should provide and what SR (may be an application client) is supposed to get. The *Pre-condition* and *Post-condition* elements are separately modeled as a list of values and their types, considering each value as a condition. The type information is necessary to validate and execute the processes. The *Signature* part includes information that is unique to the SP. Because this information is different from one SP to another, the model needs to have different identifiers and values for specifying signature fields. Also, each identifier needs to have a defined data type. This is because *Signature* information is needed when the SP enters into a contract for service execution. The type of data, in any element of service or contract, can be either a primitive or complex type. All the examples used in this paper involve only simple types, although any complex type defined by SP can be included in the model. From modeling perspective, each signature stores a list of identifiers, and their values with types. The *Result* attribute stores the information returned after service execution. Although service execution is in layer AL, which is not within the scope of this paper, we include this attribute for completeness of database structure. We model *Results* by a list of keys (identifiers) and values with types. An identifier represents the name of the returned value. In the element SA includes the specific characteristics of a service that add another dimension to describe certain service aspects in more detail. As an example, to use a diabetic body-ware sensor, it is necessary to state its *safety* (risk) factors, the body locations where they can be used, and other technical information for connecting it with other “things” (devices). The attributes *Name*, *Value*, and *Type* of SA are to specify additional information that characterize the service functionality. We remark that the *Type* field is needed to enable interoperability in the system. The element SNF includes essential properties that might promote service functionality acceptance. A simple example is *cost* of a diabetic body sensor. The four attributes *Name*, *Value*, *Type*, and *Description* are necessary to model a property in SNF. Although the syntax of SNF and SA look similar, they have different semantics, and serve different purposes. Both taken together enrich the description of service function.

### 3.2. Modeling Contract Part of “Things”

The *Contract* part in a *Thing Model* encompasses all the information that are changeable with *contexts* of service request, service composition, and service delivery. By fixing the service part we can change the *Contract* part and create different *Thing Model*. A primary advantage of this model is that the contract part can include legal and contextual constraints that are normally different in different locations. This modeling feature enables the IoT layer AL to modify just the contract part of a service and provide services with same functionality but with different contexts to different clients in different regions. For example, an implanted sensor could be used in different countries such as Canada and England. Because, every country has its own restrictions and laws on some services, every country needs to have a different contract. For this particular service, the age restriction for installing a sensor on a body may vary depending on the country. By including this information in *Contract* we create different contracts to suit the legalities of different countries. The contract part in our model includes the three parameters *Trustworthiness*, *Legal Issues*, and *Context* that specifies *ContextInfo*, *ContextValue*, and *ContextRule*. An example of context, written in the syntax [10], is “[Place:Canada,Time:{after2001},Policy:Body−Ware]”. This context might qualify when and where a policy on body-ware sensor is to be applied. A context rule, written in predicate logic, specifies the constraint to be satisfied in the context of service delivery. It is possible to define context variables and for each variable substitute a context value at the instance of service provision. The *Trustworthiness* parameter stores trust information related to service and/or service provider. Thus, *Trustworthiness* is composed of *Service Trust* and *Provider Trust*. *Service Trust* includes information that is related to service quality, such as safety. In general, *Service Trust* may be composed of safety, security, reliability, and availability attributes. *Provider Trust* lists the claims of the SP in some quantitative or qualitative terms. *Provider Trust* includes trust recommendations of peers and reviews of clients. However, because of the generic nature of trust information, we decided to model both *Service Trust* and *Provider Trust* as a list of names, values of those names, and their data type. *LegalIssues* are related to the business model and trade laws in the locations where services are made available. To model *LegalIssues* part, it is both necessary and sufficient to have the ability to store two string values for each rule that could be retrieved and compared to other string values. The first value is called *Informal Rule*, which is a textual representation of the rule to be read by service requesters. The second value is called *Formal Rule*, which formally represents the rule to be used by the system. It is assumed that the service provider enters both values during service publication.

**Example** **1.***Figure 3 illustrate our modeling principles for the example described below. Let* sense-sugar *be the function name in a diabetic sensor. A pre-condition of this function can be:*
Pre-Cond:validID(PatientID,DeviceID)∧Diabetic(Status)
*where validID and Diabetic are functions used to validate the IDs of the Patient wearing the Device, and Diabetic validates the health status of the patient who wears the device. A post-condition for the function can be*(sugar-Value=x.y)∧(4≤x≤12)∧(0≤y≤9)
*where x.y is the glucose level read by the sensor. Let the Signature information be (Code=(XXX,string), and the Result be (ValueStored=T). The* Signature *is to save rights for the patient and the service provider, and the* Result *guarantees the functionality of the service. The non-functional property price for the service can be:*
Price:(value:150$perDay),(type:Real),description[(CurrencyType=USDollar)].
*For this example both service and product (devise) attributes should be modeled. Some of the product attributes that are of interest to the consumer can be Company:[xxx,string], Color:[Grey,string], Year:[2016,Integer], Model:[zzz,string], stripsType:[yyy,string],Risk:[Low]. The name of the attribute is the identifier which is mapped to its value and its type. The service context includes information of the service provider SP and service requester SR. The SP provides a rule stating that the service should be provided to a person who is at least 18 years old. This context rule is represented as follows:*
(PatientAge≥18)∧(Status=Diabetic)∧(PCity=HosCity)
*The service provider context below includes meta information of the SP.*
SPC=[SPID:a124,Location:NewYork,Name:ABC,Address:15,Fifthavenue]
*The SR context below prescribes the context to be satisfied by the SR for proper service delivery.*
$$\begin{array}{ll} C= & [Name:Smith,\;PID:123,\;Age:\geq 31,\;Location:New\;York,\;Address:25,\;Third\;avenue,\\ & \;Status:Diabetic_{T}1,\;AppID:abc] \end{array}$$
*The* trustworthiness *property of a device can be written in terms of* ProductTrust *and* ProviderTrust *as:*
**ProductTrust**
$$\begin{array}{c} {}[(Safety:)\;(value:riskFree),(type:string),\\ (Security:)\;(value:PID\;protected),(type:string),\\ (Availability:)\;(value:24\;hours\;advance),(type:string),\\ \left(Reliability:)\;(value:no\;defect\;record),(type:string)\right] \end{array}$$

**ProviderTrust**
$$\begin{array}{c} {}[(Rating:)\;(rank:8/10),(type:enumerate),\\ \left(Recommendation:)\;(value:High),(type:string\right)\\ \left(Certificate:)\;(value:BBB),(type:string)\right] \end{array}$$
*Two examples of legal issues governing medical device rental are given below, each represented as a rule of the form: IF〈condition〉THEN〈action〉.*
$$\begin{array}{c} \mathrm{Damage}\mathrm{and}\mathrm{Liability}:\\ (InformalValue:\mathrm{Health}\;\mathrm{risk}\;\mathrm{is}\;\mathrm{not}\;\mathrm{covered}\\ \mathrm{The}\;\mathrm{client}\;\mathrm{CrditCard}\;\mathrm{is}\;\mathrm{charged}\;\mathrm{for}\;\mathrm{new}\;\mathrm{implant})\\ (FormalValue:IF\langle HealthRisk)\rangle \\ THEN(Coverage=Null\wedge Liability=Null\wedge Charge(CrditCard))\\ \mathrm{Error}:\\ (InformalValue:\mathrm{The}\;\mathrm{sensor}\;\mathrm{is}\;\mathrm{not}\;\mathrm{working},\;\mathrm{if}\;a\;\mathrm{sensor}\;\mathrm{returns}\;\mathrm{null}\;\mathrm{or}\;\mathrm{error}\;\mathrm{value}\\ \mathrm{then}\;\mathrm{the}\;\mathrm{client}\;\mathrm{is}\;\mathrm{provided}\;\mathrm{with}\;a\;\mathrm{free}\;\mathrm{implant})\\ (FormalValue:IF\langle Error\rangle \\ THEN(SensorCoverage=newSensor(replace)\wedge Charge=Null)) \end{array}$$


### 3.3. Choice of Database

Because of the V-characterization of BD and the fact that not all attributes in the “things” model (Figure 1) may arise for the “things” in an application, we can easily rule that the standard traditional database systems (SQL-based) are not suitable for smart city applications. This does not imply that any NoSQL model is a suitable choice, because we need a justifiable comparison between the NoSQL databases to choose the most suitable that can handle all the Vs that affect the IoT development layers. Some recent works [16,18,21] have reported their experiences in using Cloud platform, and edge/fog technologies for specialized applications. As pointed out in [21] there are some advantages and several advantages to using Cloud. In smart city development the supervisory system in the city layer must have administrative role on the infrastructure, security, privacy, definition of contexts, and context constraints. It is not sufficient for the city supervisory system to be able to provide data and manage services and applications. Moreover, as pointed out in [22] each actor in the Triple Helix” mechanism in smart cities has a specific role to play. The role of academic research is to address every fundamental aspect of the smart city, and explore and innovate on deep content analysis of published documents. Motivated from this perspective we compare the merits of three NoSQL databases with respect to meeting the challenges raised by the 10 Vs governing the IoT layers. NoSQL technologies fall under the three categories *Key-Value*, *Column-Oriented*, and *Document-Oriented*. Each of these NoSQL technologies has many platforms to support its operations. Therefore, we decided to choose one NoSQL platform from each category for comparison with respect to the Vs. The selected databases are Redis for Key-Value [23], Hbase for Column-oriented [24], and MongoDB for Document-Oriented [25].

Redis is considered an advanced key-value database for providing five possible data structures for the value type. These data structures are *String*, *Set, Hash, List*, and *Sorted Set*. Using Redis in BD integration process to implement smart city applications is a non-trivial task. The main reasons are that it does not support aggregation and the key-value stores does not allow constructing one rich object wrapped in a table in an efficient manner [23]. The number of attribute-value pairs handled by *Hash*, *Set*, and *List* types is fixed at 2${}^{32}$−1. Although this upper limit is a huge number, the volume of BD is currently in the order of terabyte and is rapidly expected to exceed petabytes. Therefore, Redis cannot handle *Volume* characteristic of BD. “Data Type Selection” for BD integration becomes inefficient in Redis because we are forced to choose the appropriate Redis data type by examining the nature of included data and query types on this information. This limits the heterogeneity (*Variety*) management in BD. Redis does not support secondary keys. To process a query with secondary keys, either we search on each secondary key and combine their outputs or create additional mappings from the generic model to keys and store their key-value pairs in Redis database. To understand this later method, assume a query mentions Email and Phone_Number. The map Email→ID maps an email to the ID of the owner of the email. We can store these key-value pairs in Redis and use it when the user requests a search based on email. Similar mappings are created for other secondary keys. Consequently, we have to store all possible maps of secondary key to the respective primary keys and store them in Redis database in order to allow the user to search the data based on secondary keys. This will create additional storage and overhead in searching time. The data size in BD is already huge and by adding such mappings to store secondary keys would be a nightmare that deteriorates the management of not only data *Volume* but also *Velocity* and *Vincularity*. In particular, linking process becomes more challenging and time consuming, and thereby Volatility increases in BD management because it becomes necessary to specify how long this additional information should remain in the storage. Because this new data is related to what already exists it is hard to predict its life span, which in turn affects the (*Veracity*) characteristic of BD. Based on this analysis we decided not to use Redis for smart city applications.

Hbase is designed to work with massive data by storing data in tables which are not similar to traditional tables of SQL databases. A table in Hbase is the big table that can expand vertically and horizontally, i.e., Hbase design allows increasing the number of rows and columns. The columns are like variables assigned for each row. Although Hbase supports the flexibility to provide different columns for each row, it has the following limitations.
*Limited Column Family:* The columns of Hbase are grouped by *Column Families* (*CFs*). Hbase requires the number of CFs to be small. It is better to keep a maximum of three CFs to optimize the performance. If it is necessary to have more than three CFs, it is better not to query more than three at any one time. Because the description of “things” is generic (see Figure 2), and BD applications need responses to complex queries directed at data integrated from different data sources, ideally there should be no limitation on the number of CFs. To be consistent without modeling goal, we want to provide the ability to include as many attributes as are necessary to describe the “thing” in a query. With restrictions, it becomes complex and inefficient to find and rank the “things” to requesters based on their preferences of different attributes. Additionally, Hbase limitation on CF causes an obstacle to achieve BD goals. To be convinced from real-life situations, consider the analysis goal “ find the patient group who take the same set of medications, who live in the same neighborhood and use the same set of clinical facilities”. Clearly the Hbase restriction on CF will make this analysis inefficient.*Column Qualifiers:* In Hbase there is no limitation on the number of column qualifiers. It is better to use column fields as stored information because this increases efficiency. However, Hbase stores the column qualifier key while not limiting the number of column qualifiers. Hence, creating long column qualifiers can be quite costly in terms of storage. This could be an issue in our case as the thing structure is quite rich and creating a qualifier could result in costly storage.

In addition, Hbase is not suitable for transactional applications that require RDBMS features such as transaction, triggers, and complex query operations. In smart city applications, the “things” database has to support real-time processing ability and ad-hoc querying. Therefore, we conclude that Hbase cannot provide support to smart city development.

It is claimed [26] that MongoDB is the leading non-relational database for BD management, when compared with different 451 NoSQL databases. It is an open source document-oriented NoSQL database, in which a document is a record made up of a group of fields and their associated values. The number of fields need not be the same in all documents, i.e., each record can have a structure different from the structure of another record. MongoDB is supported by ad-hoc query system that allows querying by a specific field of a document. It supports aggregation operation which is beneficial to using map reduce methodology. One of the most important features of MongoDB is its indexing facility. Aside from the mandatory indexing that is automatically done by MongoDB *System_id*, a secondary indexing facility in MongoDB enables adding indexes to other fields of the documents. The following features [26] seem sufficient to tame the V-model complexities of BD.
*Veracity, Validity, Value:* MongoDB’s distributed architecture separates the analysis processes, such as duplicate detection and machine learning, from querying processes. This helps avoiding long ETL (Extract, Transform, and Load) processes from impacting the operational application, and keeping the data more reliable in its source. As a result, of achieving high *Veracity* at early level, both *Validity* and *Value* of integrated data are enhanced.*Volume and Variety:* Variety not only refers to the different schema integrated from different sources but also to the flexibility in merging them into a single schema without losing any data. The semi-structured schemas are managed in MongoDB “*without giving up sophisticated multi-record ACID properties of transactions, schema governance, data access, and rich indexing functionality"*. This helps to manage both *Variety and Veracity* characteristics of BD. MongoDB provides support for highly scalable data management for geographically distributed data centers and cloud regions with high level of availability. This basically supports the management of *Volume* characteristic of BD.*Vincularity, Velocity, Validity, Volatility:* Indexing at multiple levels enable linking of related data at different sources (“things”). Hence, *Vincularity* characteristic is maintained efficiently. After data integration, validation process could be enhanced through secondary indexes. “Streaming Data Pipeline” feature in MongoDB allows developers to build reactive, real-time apps that can view, filter, and act on data changes as they occur in the database. Combining this feature with the ability to reduction of latency and fast data execution for queries, both *Velocity* and *Volatility* characteristic of high data streaming can be managed.*Visualization:* MongoDB uses different methods and visualization tools such as Tableau to efficiently access and display data it stores using standard SQL.

Other attractive features of MongoDB include its access control and encryption of data which can be enforced both at compile-time and at run-time, and advanced services of Cloud. Because all the Vs of BD are satisfactorily handled in MangoDB we were encouraged to experiment with MongoDB in the smart city development architecture discussed in Section 4.

### 3.4. Things Database: MongoDB Implementation

To enhance and optimize “things” usability and efficiency in different contexts of applications, knowledge about the “things”, the services they provide and their interrelatedness should be stored. To achieve efficiency in retrieving stored knowledge of different domains, based on functionality and non-functional aspects in the contract part of “things”, “things” may be clustered. Figure 4 shows how clustering can be organized in MongoDB using domains and sub-domains. A cluster is a collection of documents. Using MongoDB features we have implemented the three collections *Domain Knowledge Collection* (DKN), *Provider Collection*, and *Things Collection*, assuming that most frequent queries will require “things” from one or more of these clusters.

The DKN collection includes a single document for each domain, sub-domain, and function, as shown in Figure 5. A strong feature of MongoDB is that it allows different documents within a collection to be structured differently. Exploiting this feature, we structure domain document, sub-domain document, and function document differently by mapping structure of the three first levels of generic model into MongoDB representation.

The domain document is structured to have a direct one-one mapping from the domain part in the generic model. Notice that both *Service Provider Context* and *Service Requester Context* fields are embedded documents. The advantage is that these sub-documents can be retrieved and used separately from the domain document. The rest of the fields in domain document are either records or atomic values. In sub-domain, document and function document fields are either records or atomic values. The mapping from the generic model preserves consistencies between parents and children nodes.

The provider collection is a set of documents, where each document stores knowledge of one service provider defined in the generic model. The document structure for a service provider is shown in Figure 6. In the field *Followers*, the identities of all provided services are wrapped. The field *Followings* refers to the parent node of the service, which defines the function used by this provider.

For “things” collection, it is decided to store them in separate collections. Each collection represents the service function that the *thing* provides, as shown in Figure 7. The collections are named by the function IDs. A thing in the generic model is represented in three documents stored in the collection to which the thing belongs. These documents are used to represent service, contract, and context information defined in the generic model. Because many “things” can have the same functionality, we assign the function ID to name the thing that stores all “things” with this functionality. The advantages of this design include (1) the ability to modify context parts of a thing, either independently or jointly, without affecting the service part, and (2) enable the system to automatically find all “things” that belong to the function selected by a service, by finding the collection distinguished by the function ID. The three documents in the collection of “things” are linked through the fields *Followings* and *Followers*. These fields enable tracking of the contract and context information of a service. The context document is richer than the other two documents, because we need many embedded documents in it. The *ContextInfo* part of a thing contract is mapped to an embedded document that contains all dimensions as fields with their types as values. The *ContextRule* is a defined field in the contract that includes a string value representing the rule entered by Service Provider. The *ContextValue* is mapped to an embedded document that contains fields and arrays. The value of attribute date/time, clientID, providerID, and serviceID within the embedded document of *ContextValue* are atomic. Each dimension of the context is modeled as an array structure, which wraps the information specific to each dimension in one memory block. Thus, all information regarding one dimension including sourceID, date/time of collection, and value of the dimension can be retrieved by the name of the dimension. The rationale for representing dimensions as arrays instead of embedded documents is to reduce the number of levels of document embedding. Increasing the number of levels of document embedding requires complex query processing and retrieval, which makes MongoDB operations to be resource intensive. Thus, when an update operation is performed, only the last updated field and the values of dimensions are updated with a single query.

### 3.5. Data Integration

Two essential steps in smart city development are data integration and analysis of system smartness. In this section, we propose a top-down data integration process that is driven by analysis goals. The three layers in Figure 8 are (1) the bottom layer that models the physical Smart City network of applications which feed data to databases, (2) the top layer that models the goal-oriented analysis leading to the required targets for analysis, and (3) the middle layer that models the role of database management for data integration and feeding the targets. The goal, shown at the top, is to analyze some specific aspect in Smart City application. In general, the chosen goal has several sub-goals each of which will require investigation of a specific target. To answer the queries arising from sub-goals and reason with it, we need to integrate the data from those application domains and feed the targets. This data integration process is shown in the middle layer. The database management system, shown in the middle layer, links the data stores of “things” and other external sources in the lowest layer with the integration process in the middle layer. It oversees data integration and facilitates the query processing at the target of each sub-goal.

### 3.6. An Example of Data Integration in MongoDB

The two kinds of integration are *Virtual Schema Integration* (VSI: Data Federation) and *Data Consolidation (ETL)*. The VSI model is specifically built for each target and the resulting virtual schema includes the data sources selected to feed the target with the required data. Thus, it fits our goal-oriented data integration model (Figure 8), and helps us to manage the number of data sources necessary to integrate for a query response and hence results in better BD management. For example, for “diabetic type1” patients, the insulin amount per meal is affected by the amount of carbohydrate a patient takes per meal. However, when the patient exercises the sugar level in the body may be lowered. This hypothesis needs to be verified to conclude that exercise is a factor that affects the insulin amount in the body of patients. Therefore, it is important to gather data for diabetic people to find the pattern to adjust the insulin amount to consider not only the carbohydrate factor but also exercise hours per day.

Assume that a Smart City application automatically monitors the health status of seniors and offers them health care services online. As part of this broader system, we consider a small subsystem that monitors the exercise activities of seniors and investigates how it impacts the glucose level of patients. In this subsystem, there are two applications, one that registers sugar levels in a patient (Diabetic Application: Figure 9) body with every meal, and another that registers the exercise durations (Exercise Application: Figure 10) per day. Diabetic application registers periodically the readings of sugar level every day and stores the results in a set of pairs of the form (time, sugar level). The date is used as a secondary indexing attribute in MongoDB. Exercise application registers the exercise type done every day. In the exercise application, the date is also considered to be a secondary index attribute. Exercises could be stored in different ways. For example, it can be an array of pairs which include the duration spent on a specific exercise and the type of the exercises applied on that time. The calories could be given either individually with every exercise or given as a cumulative value. The applications could be structured in different ways in MongoDB; however, this is not an issue as MongoDB provides different functionalities and features such as aggregation that helps manage heterogeneity of the data schema.

Both applications monitor the information of one patient using implanted sensors “things”. The integration of these two applications are required to investigate how exercises affect the sugar level in the patient body. The integration process for this example is illustrated in Figure 11. As shown in the figure, the only target is “target 1”, and the two application databases should be integrated to achieve this sub-goal. If both “public general key and ontology” are provided by the domain experts, the schemas of the two application databases are merged to obtain the schema of the integrated database (for target 1). The result after integration is shown in Figure 12.

The public key helps in identifying the records and simplify the process of linking records together, i.e., the ontology helps in understanding the semantic of attributes in order to merge the ones with the same meaning and to choose the term that is preferred in the integrated schema. From the description of data profile of both applications, we find that each application has one collection that starts with a profile document. The profile description in both applications has the attribute *ID: 1*, which refers to one member in the Smart City. From other parts of database, we found that both applications use attribute *Date* to indicate the time when the data was registered. Using this as secondary attribute, the data generated on the same date are merged.

It is possible from the integrated database to query and find how exercise affects sugar levels. However, to conduct a rigorous inference, we need a reasoning capability embedded in smart city system. Domain expert knowledge, rules for inference, and data from the integrated database (considered to be facts) are to be part of the inference system. A knowledge base system is necessary to infer certain critical properties on which diagnosis and decisions can be made. Some of these issues are studied in [11].

### 3.7. Experimental Observations–Comparing NoSQL Databases

In Section 3.3 we gave a qualitative comparison of NoSQL databases with respect to the V model of BD and justified the suitability of MongoDB for smart city design. In this section, we explain our experiences [27], and the experiences reported by many others [28,29,30,31,32] in using *Yahoo! Cloud Serving Benchmarking tool* (YCSB) [33] to test the performance of NoSQL databases. The comparative study [27] was done to justify the choice of MongoDB’s for managing and querying *configured services*, which are “complex things” in Service-oriented Systems management. Since we have modeled a “thing” as a service with contract, which is a service-oriented model of a “thing”, we are quite justified in using our observations in this current study. In addition, the MongoDB choice of ours is very well supported by the findings of the experiments done by the rest of researchers.

(YCSB) [33] is a tool to compare the relative performance of NoSQL database management systems, and is known as “BD benchmark”. Since BD storage management requirements for smart city design includes performance, throughput, scalability, availability, and stability, it is hard to obtain a large volume of real-life data to test all these requirements. As observed in [30,31,32], YCSB can be used to assess the capabilities of NoSQL databases with respect to specific requirements.

YCSB provides six workloads. Each workload performs different combination of operations on different ratios. Table 1 shows the six workloads defined by YCSB, where the sizes of the respective workloads are 10,000, 100,000, 500,000, and 1000,000. In [27] we executed these operations three times for each workload and plotted the average of results in Figure 13, Figure 14, Figure 15, Figure 16, Figure 17, Figure 18 and Figure 19. For a fair examination, all experiments were performed on the same machine. Our observations and those of others are summarized below.

#### Observations

In Redis, a single operation on a complex structure will be translated to several operations. This is because Redis does not support table or document structure. It is observed in experiments Redis was not able to meet the request of insertion of 600,000 records at once [30], although it was better for managing rapidly changing data [32]. Although it might show good execution performance for operations on simple structures, when the structure of data is complex the performance is slowed down. With our experience in trying Hbase we agree with the remark [34] that “Hbase was quite challenge for us. The terminology can be deceptively reassuring, and the installation and configuration are not for the faint of heart.” Indeed, we found that understanding and dealing with Hbase was not easy. There was much time spent to understand how to structure tables, how to configure it on a machine, and what structural characteristics are necessary that might lead to maximize Hbase potential. Although each table in Hbase can have more than one column family, its performance degrades if the table has more than three CFs. This does not help when structuring rich components. It is reported [30] that Hbase has the best data loading; however, it has the worst execution time and lower performance when compared among Column Family databases. With MongoDB we found that its features and characteristics help with structuring “Things” Database. The ability to query data and fields easily meets our database requirements in query system. Also, the ability to store different data type such as arrays and sub-document are very useful in structuring the clusters of things hierarchy *DKN, providers*, and *Things* (*Service, Contract*, and *Context*)). In addition, MongoDB supports dynamic document expansion which is an essential feature to provide for “Things Database”. With this feature “things providers” get the ability to enhance their services by updating service description, adding new attributes, and adding new data. Some analysis is required to find the proper structure to include within SP records to facilitate dynamic expansion, which might affect the query processing performance and consistency of data. It is best in handling dynamic and frequently written data, and for these reasons it has been used in two healthcare data management studies [32]. Averaged over different workloads, MongoDB has a stable performance. Based on these observations arising out of our experimental study in using YCSB, and in the light of the previously discussed V-model comparisons Section 3.3, we identify the following five important features to prioritize for choosing “things” database.
*Stable Performance:* The performance of a database in YCSB testing shows stable behavior over different workloads.*Indexing:* The database should provide indexing facility for fields of records and/or the document.*Fields Querying:* The database operation should give the ability to query specific fields of the record and/or the document.*Hierarchical Structure:* This is necessary to support embedding or linking among entities.*Easy Usage:* The database must have an easy-to-use user interface ease that allows configuration, installation, and coding.

Table 2 shows the presence/absence of each feature and the extent to which each feature persists in each NoSQL. Our decision to choose MongoDB as the most suitable NoSQL database for implementation in smart city is based on this comprehensive comparison.

## 4. Database Support for Smart City Realization

In this section, we first review a literature survey to point out that database structures were not adequately investigated for smart city development. Some used Cloud platforms, and Edge and Fog Computing technologies [16,18,21], and many others [35,36,37,38] assumed database to exist as “black box” and focused only on context-awareness issues. Compared to these works, we next establish the significance of our work through a brief description of our current work on smart city development architecture [11].

### 4.1. Related Work

Databases are necessary components in building Smart City applications. We restrict our discussion to smart city architectures which includes, but not discuss, databases. Recently [35,37,38,39] different kinds of databases have been introduced in Smart City architecture to enhance data processing and investigative analysis. They have assumed the database support to exist as a “black box” for context-aware systems, which they view as the basis for Smart City development. Since smart city development brings in new challenges in using context-aware architectures and because of the scale and variety of the data and resources required for such applications, we feel a critical study of database design is warranted to manage both the sensitivity and criticality of information. Consequently, in our research we took up the unsolved issue of database design, and studied the role and details of databases in such systems for Smart City development.

The Context-Awareness Sub-Structure (CASS) [35] is a server-based middleware intended to support context-aware smart applications on hand-held and other small mobile devices. The architecture of CASS contains three main components, namely a sensor node, the CASS middleware which includes a database, and a hand-held computer. The CASS uses a relational database to store the data for the application, which includes context information. In [39], the author introduces the Context Broker Architecture (CoBrA), which is an agent-based context-aware architecture aimed at “intelligent spaces”. “Intelligent spaces” are physical spaces such as offices, bedrooms, and vehicles that contain devices that offer users pervasive computing services. The core component of this architecture is the context broker, which contains a “context knowledge base”, a “context reasoning engine”, a “context acquisition module”, and a “privacy management module”. The “context knowledge base” is backed by a MySQL database. Consequently, these works cannot handle BD.

The authors in [36,37,38] mention that a database is used in their architecture; however there are little or no details about the database schema, the type of information stored, and how the data is managed. In [36], the authors introduce the Context Toolkit, which uses the concept of context widget in the architecture. In this paper, the authors mention that they use a database to store historical data. The paper does not give details on the type of database used, the type of data used in the model, and how the database is managed. The work in [37] includes a data store in their Context-Aware Framework (CAF), which records information such as associations between sensors and dimensions, situation expressions, associations between situations and adaptations, and associations between reactions and actuators. There is no mention of what schema is used for this data or how the database is managed. Finally, the authors in [38] introduce an Agent-Based Context-Aware Architecture for a Smart Living Room. There are two databases present in this architecture, namely user’s preferences and knowledge base. For the user’s preferences database, a user must fill out his preferences to be used in smart living operations. Although they present examples of user preferences, it is not mentioned which schema is used, which database management system is used, and how the data is managed and by whom. The knowledge base database is also used only as a black box.

### 4.2. The Proposed Smart City Development Architecture

Our architecture, shown in Figure 20, is based on sound software engineering principles and ubiquitous computing paradigm. Integrating “things” abstraction right from requirements stage, we have included two databases, called *Application Data Store* (ADS) and *General Data Store* (GDS), in the architecture to handle “static” and “dynamic” objects and BD. They are both MongDBs, serving different purposes during the interaction of system components. The type of information stored in the GDS must be general to all applications, as the name suggests. The other data store is the ADS and is used for specific applications. It stores facts, rules, and context information related to that application. Facts are generally static and can include information such as a user information, user preferences, and privacy policies. The rules that are stored in the ADS are predefined adaptation rules which together with context is used to determine the appropriate adaptations. Finally, the context information stored in ADS is dynamic and generally comes from the sensors and other input devices in the environment. The principal components *Sensor Component* (SC), *Context Component* (CC), *Inference Component* (IC), and *Adaptation Component* (AC) interact with ADS and GDS components. The collaborative actions of the components can be conceptualized into *Input*, *Process*, and *Output* stages. The roles played by SC in the input stage, CC and IC in the process stage, and AC in the output stage contribute to overall smartness. During the input stage, the controllers to database interfaces handle large *Volume* of a *Variety* of data which may arrive with varying *Velocity*. During the process stage, *Vincularity* is handled at data integration stage, *Veracity*, *Validity*, *Vitality*, and *Volatility* are handled by the process components. During the output stage, information from database are retrieved and sent to actuators to satisfy *Value*, and *Visualization* demands. Because the process components interact with the database during process stage, it is clear that the database plays a crucial role in all three stages of the Smart City system.

#### 4.2.1. Input Stage

The sensor component is the point of entry in the "input" step in the smartness steps. It is comprised of a controller, a transformer, a validating unit, a data synchronizer, and a data store connector. The controller subscribes to the data streams of sensors and other input devices, fetches device information and policies from the data stores, calls the transformer and the validator for each device reading, uses the data synchronizer to aggregate and normalize the readings, and calls the data store connector to write the new data into the ADS. The transformer uses the stored sensor information and any relevant policies retrieved from the GDS to transform the data. The validator process ensures the validity of the data using the information and policies fetched from the GDS. The data synchronizer aggregates data coming from sensors given a criterion such as type or area. It also normalizes the data before it is stored in the ADS. Finally, the data store connector uses specialized methods to connect to the defined data stores. It receives the transformed and validated data, formats it according to the data store used, and writes to the ADS. The sensor component generally requires that the sensor information be stored in the GDS beforehand. This information must include the data unit if it is available, as well as the possible range for a data reading. A data retention policy can be stored in the GDS to ensure that all data being processed by the system remain valid and are up to date. The sensor component writes the sensor readings to the ADS in real time so that they can be processed by the context component subsequently. Figure 21 shows the main interactions between the sensor controller and the data stores.

#### 4.2.2. Process Stage

The sequence diagram showing the interaction of different components with the databases is shown in Figure 22. The context component listens to changes in the sensor reading data in the ADS, and triggers when new data is added. The context component is comprised of a controller, context manager, context builder, and data store connector. The controller has a listener that monitors the sensor reading data in the ADS. When new data is added, an event is sent to the controller, forcing it to run and fetch the new sensor data from the ADS. The controller then calls the context manager and the context builder for the new readings, and calls the data store connector if the new context data needs to be stored in the ADS. The context manager manages the incoming sensor readings as well as the stored context, both of which are stored in the ADS. It keeps the stored context data up to date, decides whether to store new data as context, and requests new data if needed. The context manager is important because it controls the context information for the application, thus controlling the efficiency and accuracy of the system. The context builder uses policies stored in the GDS to ensure that the context is built in a correct way. In this architecture, the context is defined using key-value pairs of dimensions and attributes. The context builder is only called if the context manager decides that the new sensor reading is to be stored as context in the ADS. Finally, the data store connector in this component is the same one used in the sensor component. It connects to the ADS and stores the new context data in the appropriate location. The context component requires the GDS to have context policies, which include syntax, special cases, and any other useful information used to build context instances. It also assumes that the new sensor readings are stored in the ADS. This component stores context instances in the ADS so that they can be used by the inference component.

The inference component receives context information based on sensor data from the ADS and uses adaptation rules and facts loaded from the ADS to produce an action. In terms of smartness, this is the component that transforms the input to an action to be sent to the adaptation component. This contains a controller, an inference engine helper, and an inference engine. The controller continuously listens to changes in context information from the ADS. Upon receiving an event indicating that one or more context instances have been added to the ADS, the controller fetches the new context information from that data store. Then, the controller calls the inference engine helper to prepare the input to the inference engine, which includes context instances, rules, and facts. Preparation of the input includes activities such as making sure that it is syntactically correct. After all the requirements are satisfied, the inference engine helper calls the inference engine with the input. The role of the inference engine is to infer new facts and knowledge based on context information, stored policies, rules, and facts from the knowledge base. As soon as the inference engine completes its run, its output is formatted by the helper to have a uniform format regardless of the inference engine. Finally, the formatted output is sent to the adaptation component.

#### 4.2.3. Output Stage

It is responsible for determining and sending the appropriate reactions to be carried out by the actuators. The sub-components of this component include the controller, actuator, and the actuator connector. When the inference component executes, the inferred knowledge it generates is sent to the adaptation component. The adaptation controller continuously listens for the arrival of new inferred knowledge. As soon as it receives one, the actuator sub-component determines which actuators need to be activated, loads their information from the GDS, transforms the inferred action into a meaningful actuator command, and sends that information to the connector of the activated actuator. This sub-component locates the appropriate actuators, connects to them, and sends the appropriate commands to them. The adaptation component uses the GDS to load mandata related to the output devices in the environment and can optionally store the results in the ADS for further analysis. The mandata for the output devices can include information such as device ID, device type, location, possible commands, and status. Figure 23 demonstrates the interactions between the general database and the components within the “output” step for smartness.

### 4.3. Analysis

The data stores used in our architecture are active, meaning that they can send triggers and events to other components in the architecture. Most of the components in the architecture do not communicate directly with each other, instead they store their data in the ADS. This data store then sends a trigger to the appropriate component so that they can ingest the data. The only exception to that is the communication between the inference component and the adaptation component, since the inference component sends its output directly to the adaptation component.

There are many advantages for having the communication between components happen through the data store. First, the components can be designed and developed independently. The only information needed would be what type of input and output is expected. In this case, the format would be the same for all the components since the type of data store would be known beforehand. It also makes maintaining and updating components easier since the input and output are known. Second, using the data store for the communication between components allows for a more controlled communication. In other words, the data store can control whether to send a trigger to a component upon receiving data. This can be very helpful in preventing unwanted behavior from the system in case of a security compromise or data corruption. Finally, it provides a centralized place to store all the data pertaining to a specific application. Therefore, if any component would need additional data, it would know where it is and would have the appropriate connectors to fetch that data.

## 5. Conclusions

A successful realization of Smart City rests on efficient representation and management of the underlying BD that supports a multitude of “intelligent” applications that provide uninterrupted reliable information and services to different segments of population in the city. The operations of Smart City applications also generate BD which must be stored and managed for future consumption and service adaptations. Consequently, we need highly efficient database systems for managing BD. In this article, we conceptualized the hardware/software aspects of Smart City applications from the perspective of IoT and the BD generated and consumed by the “things” in IoT. This resulted in a generic model of the “things”, which we then used as a basis to identify essential features of a desired database management system to manage BD in IoT and support Smart City applications. In conclusion, we found that MongoDB system is currently the best choice for managing BD and supporting IoT applications. We justified this choice, made after a careful study and comparison of the power and the merits of existing DB technologies, in particular three NoSQL databases, by illustrating the suitability of MongoDB to efficiently handle the generic model of IoT mentioned above, with the Vs that characterize the underlying BD. This was illustrated through an example for the BD integration process, which is a necessary step for investigative analysis that arises in Smart City service provision. After reviewing the literature on the architectures for the design of Smart City applications, we observed that none of them discussed in any detail the role of databases in use and handling BD that arise in and affects Smart City applications. We believe that BD data management systems play a pivotal role in development of these applications. As such, the contributions of this article are original and significant. It is original because no previous work has discussed the database design suitable for IoT. It is significance because it identifies the MongoDB system to be a suitable choice for efficiently handle the challenges posed by the Vs that characterize BD. It also offers an effective platform for data integration task that meets the requirements in top-down query processing. Despite what we can achieve, we notice the following limitations of MongoDB in achieving the full support needed in our context of development of Smart City Applications.
Some Collections may scale down. This is because of the decision we made for clustering the “things” based on their common functionality. Some collections of “things” might have only few records in it, because not too many SPs may publish such a service. Also, when the SP of an existing “things” ceases to sustain providing a service, it may be deleted from the system. Consequently, there is a possibility that a collection may end up being empty.We may need to Iterate Collections. If there was a user request that requires some operation to be repeatedly applied to more than one thing, then we need to go through different collections to apply that operation. Consequently, substantial amount of work will have to be done to satisfy such user requests. It is also possible that maintenance operations might require going through some collections more frequently than others.

It is hard to predict when these minor limitations arise. Even when they arise, they may not persist when the Velocity of data turnover is high. Consequently, an important future work would be finding a new database schema that overcomes these limitations.

## Figures and Tables

**Figure 1 sensors-19-02430-f001:**
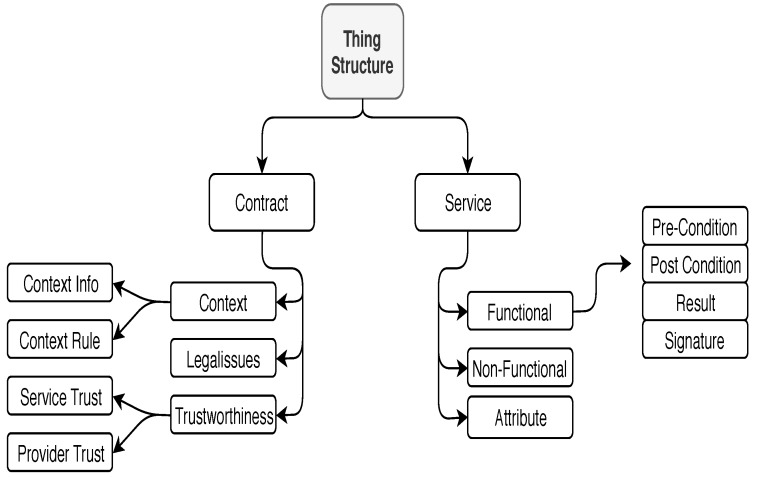
General Thing Structure.

**Figure 2 sensors-19-02430-f002:**
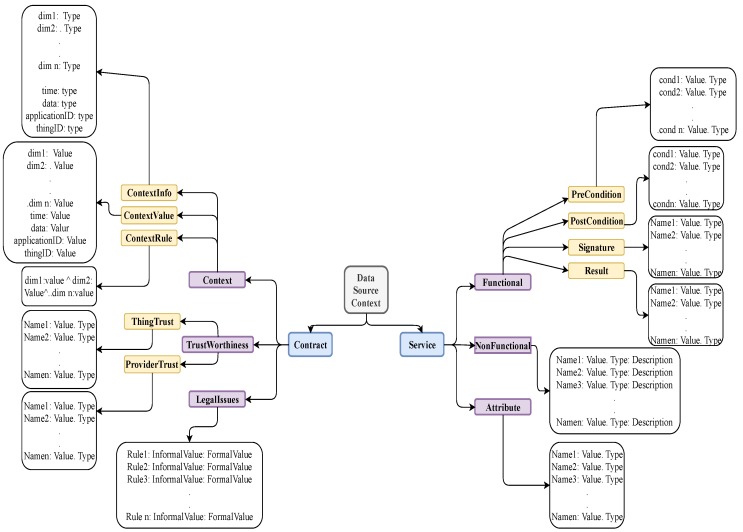
The Thing Implementation Structure.

**Figure 3 sensors-19-02430-f003:**
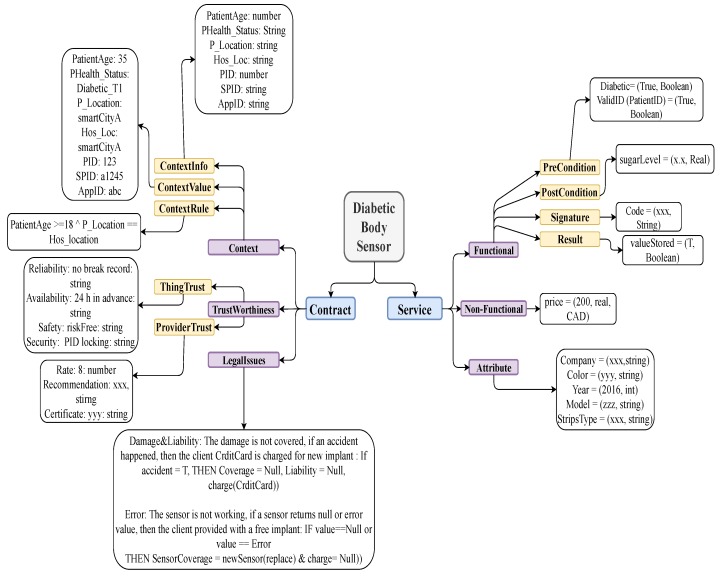
The Thing Implementation Structure Example.

**Figure 4 sensors-19-02430-f004:**
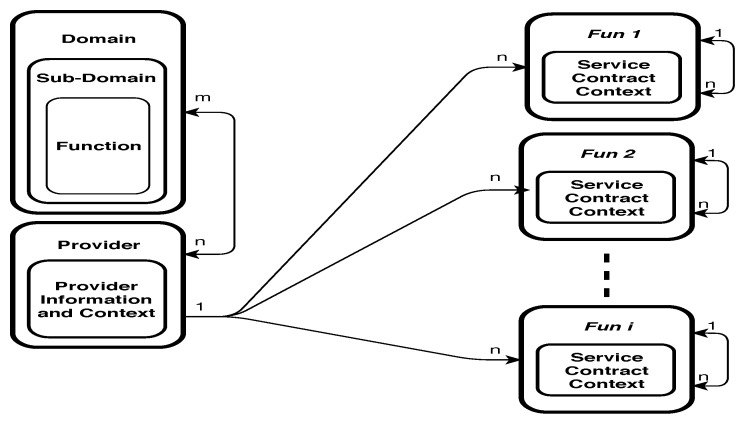
Smart City General Database of the “Things”.

**Figure 5 sensors-19-02430-f005:**
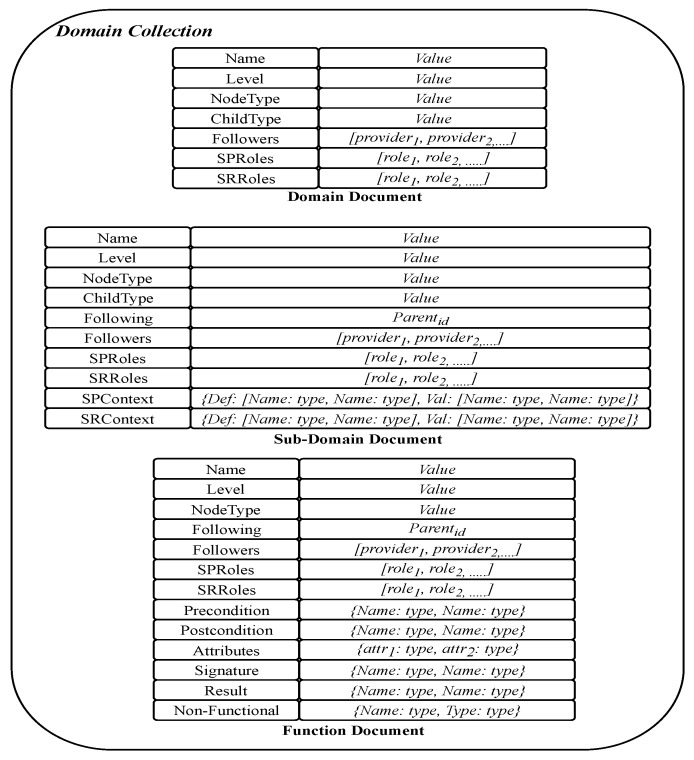
Domain Knowledge Implementation Model in MongoDB.

**Figure 6 sensors-19-02430-f006:**
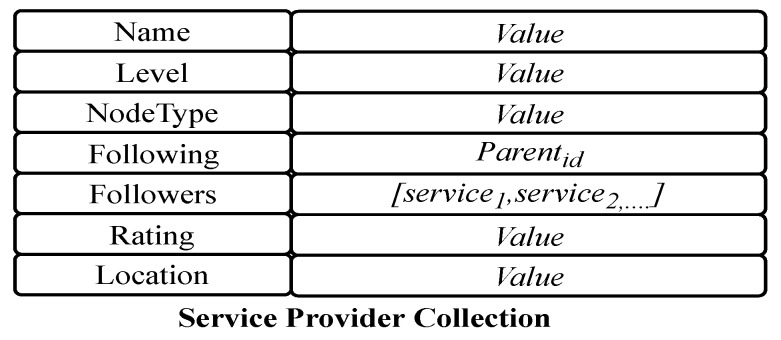
Service Provider Collection.

**Figure 7 sensors-19-02430-f007:**
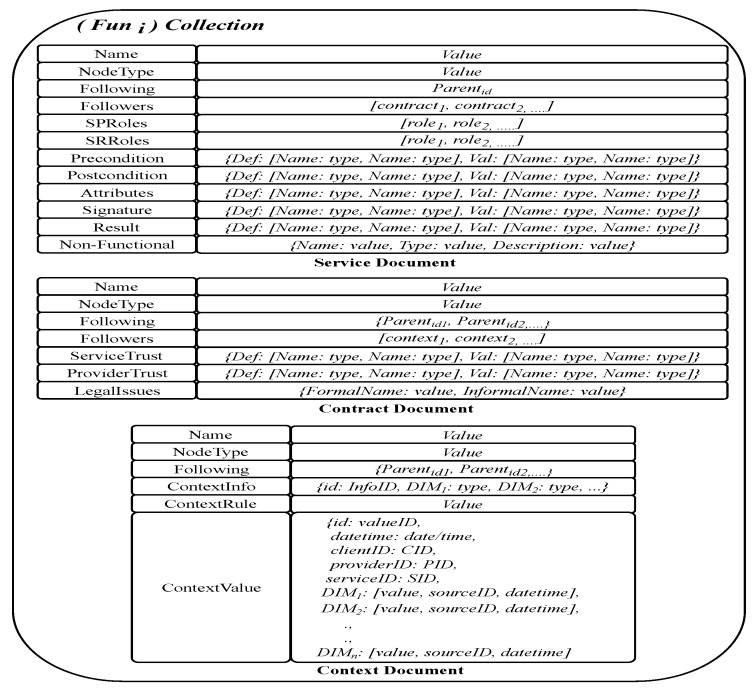
Things Collection Clustered by Function.

**Figure 8 sensors-19-02430-f008:**
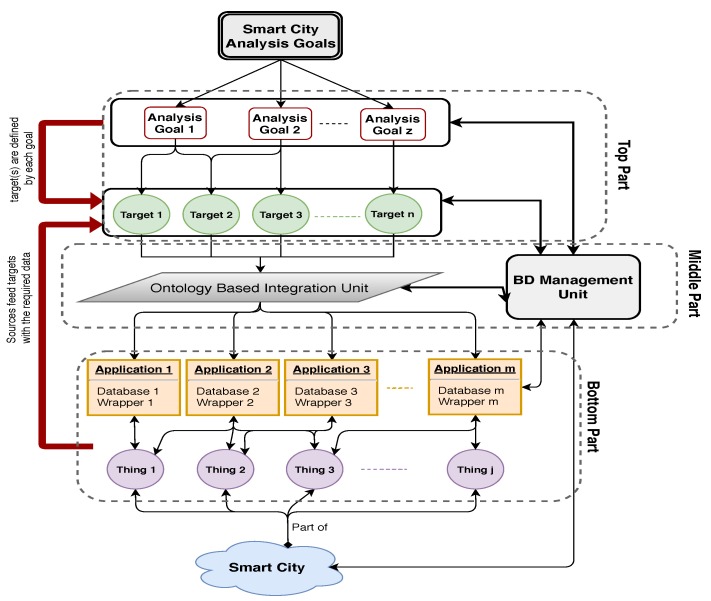
Goal-Oriented Database Integration Model for Big Data of Smart City.

**Figure 9 sensors-19-02430-f009:**
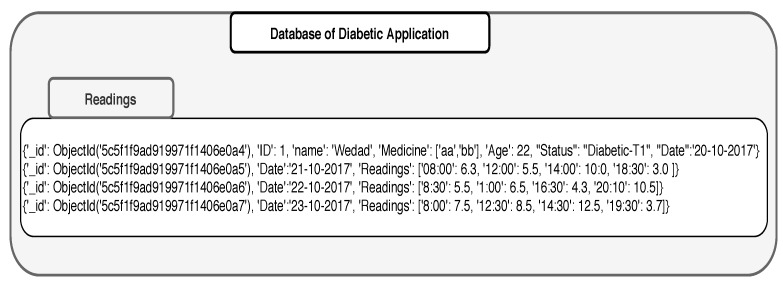
Database of Diabetic Application.

**Figure 10 sensors-19-02430-f010:**
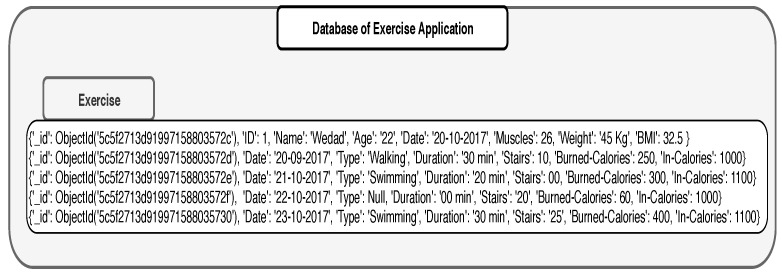
Database of Exercise Application.

**Figure 11 sensors-19-02430-f011:**
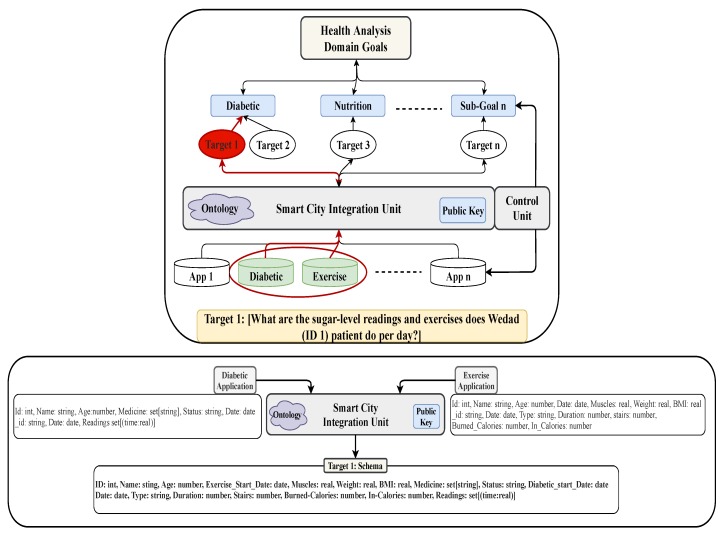
Applying Top-down Model for Integrating Diabetic and Exercise Applications.

**Figure 12 sensors-19-02430-f012:**
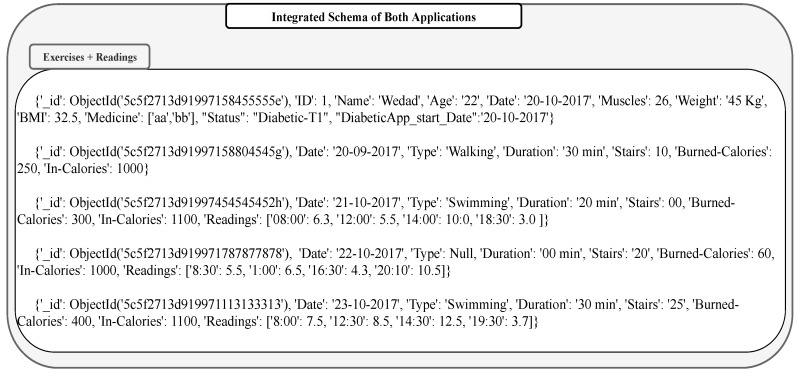
Integrated Result of Diabetic and Exercise Applications.

**Figure 13 sensors-19-02430-f013:**
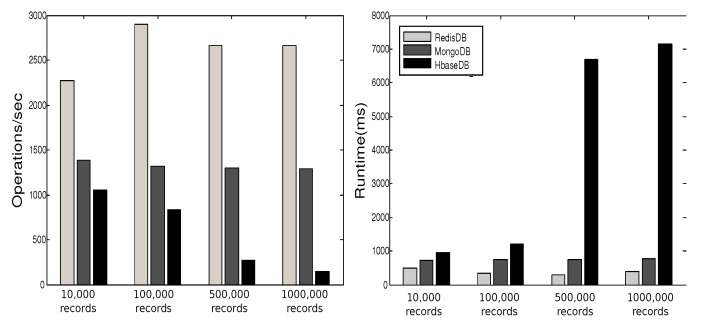
(A) Read/update ratio: 50/50.

**Figure 14 sensors-19-02430-f014:**
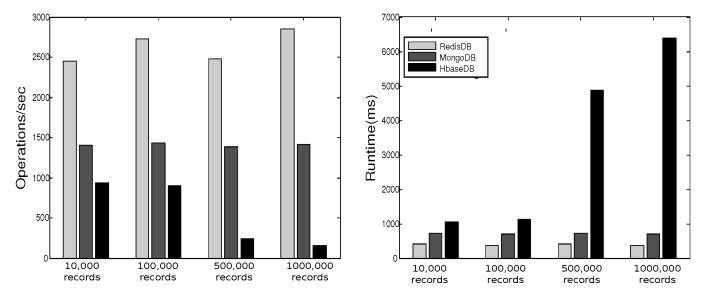
(B) Read/update ratio: 95/5.

**Figure 15 sensors-19-02430-f015:**
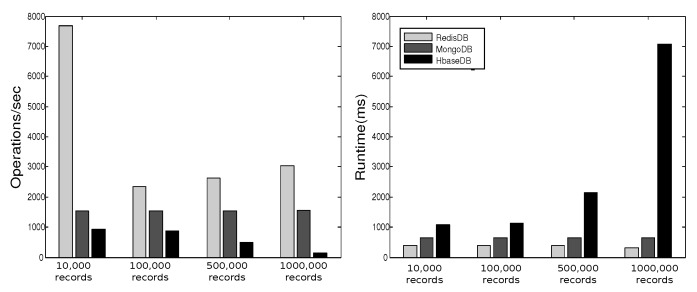
(C) Read/update ratio: 100/0.

**Figure 16 sensors-19-02430-f016:**
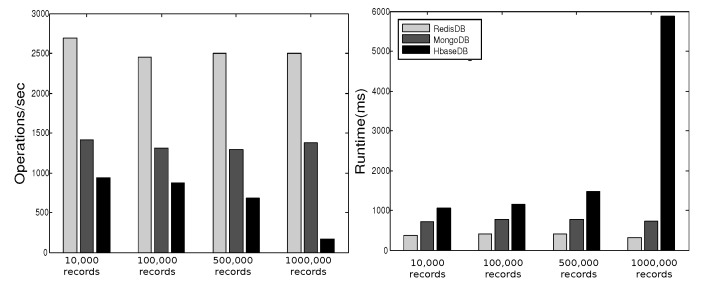
(D) Read/update/insert ratio: 95/0/5.

**Figure 17 sensors-19-02430-f017:**
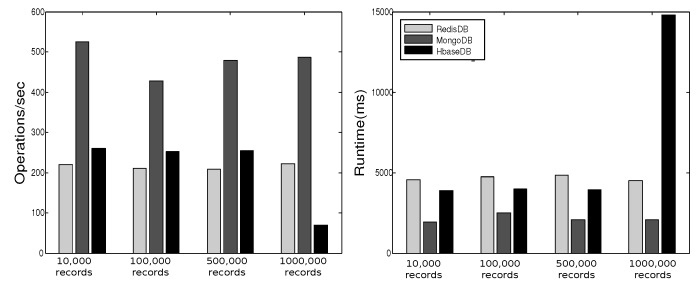
(E) Scan/insert ratio: 95/5.

**Figure 18 sensors-19-02430-f018:**
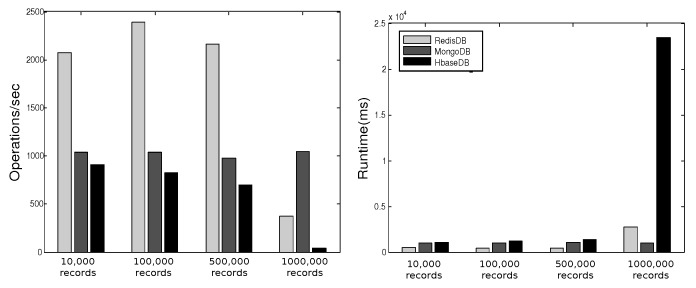
(F) Read/Read-Update ratio: 50/50.

**Figure 19 sensors-19-02430-f019:**
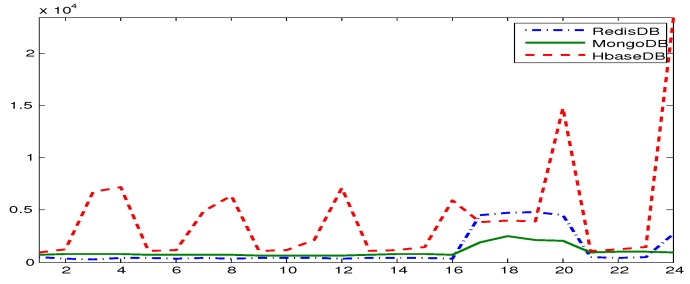
Overall Performance For all Workloads.

**Figure 20 sensors-19-02430-f020:**
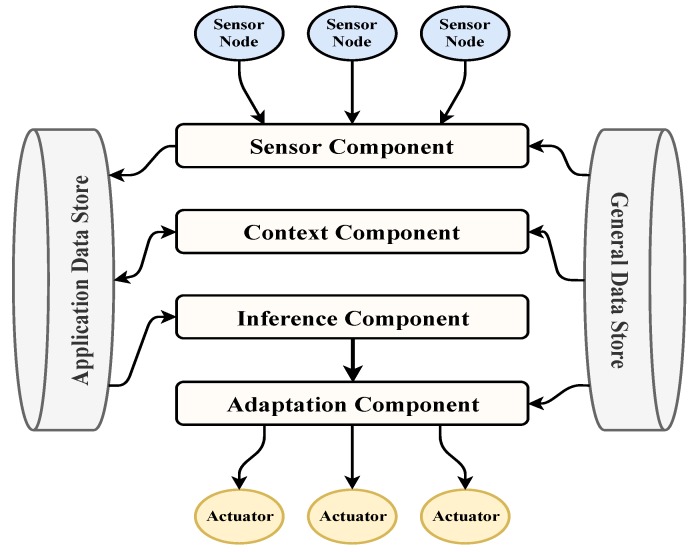
Context-Aware Architecture for Smart Cities.

**Figure 21 sensors-19-02430-f021:**
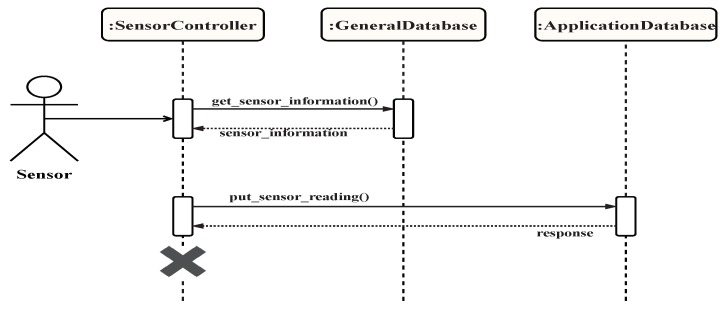
Input Step Database Interaction Sequence Diagram.

**Figure 22 sensors-19-02430-f022:**
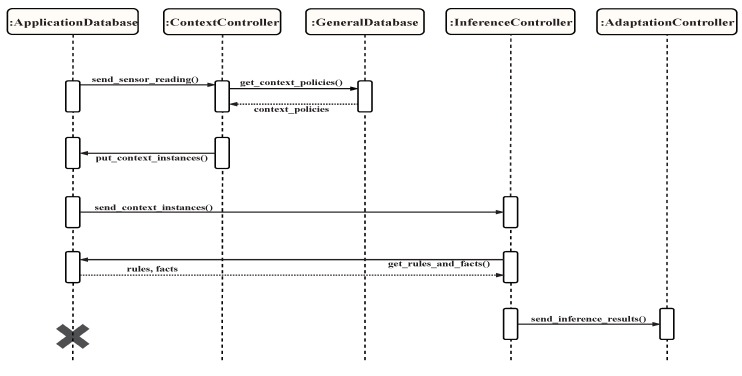
Process Step Database Interaction Sequence Diagram.

**Figure 23 sensors-19-02430-f023:**
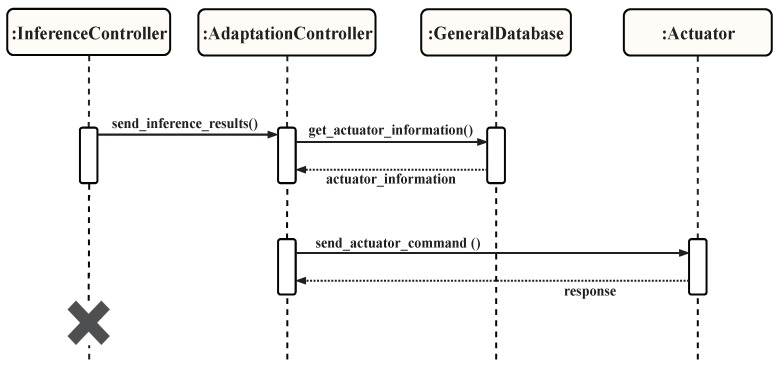
Output Step Database Interaction Sequence Diagram.

**Table 1 sensors-19-02430-t001:** The six workloads defined by YCSB.

Workload	Operations Combination	Ratio
(A)	Read/Update	50:50
(B)	Read/update	95:5
(C)	Read/update	100:0
(D)	Read/update/insert	95:0:5
(E)	Scan/insert	95:5
(F)	Read/Read-Update	50/50

**Table 2 sensors-19-02430-t002:** Comparison of NoSQL Databases.

	Redis	MongoDB	Hbase
Stable Performance	Yes	Yes	No
Indexing	No	Yes	No
Fields Querying	Partial	Partial	Yes
Hierarchical Structure	No	Yes	Partial
Usage easiness	Yes	Yes	Partial

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
