# Peer review of "Characterization and Efficient Management of Big Data in IoT-Driven Smart City Development"

_sensors, 2019, doi:10.3390/s19112430_

Round 1
Reviewer 1 Report
This is a comprehensive study by the authors to efficiently integrate Big Data into IoT applications. The study focuses on the context of smart cities and details examples of access to the healthcare system using the Mongo database.
It is curious that in the text only three times the word "cloud" appears marginally, when the development of the Database in my opinion is oriented to this computational paradigm even more strongly than the IoT. I think that the Cloud paradigm should have more presence in the paper, since it is really the computational model that exists behind the IoT concept, which only refers to the capacity that a device has to connect to the Internet.
In the cloud-based IoT paradigm, there is a current trend for devices to be able to pre-calculate before sending their information to the cloud. This trend is justified in the real time context by the delay that the cloud has in reacting and the intelligence that many of the IoT devices may have. One of the consequences of this scheme is the reduction of the database in the cloud. I think it would be wise for the model presented to take into account aspects such as "edge computing" or "fog computing". This would undoubtedly enrich the proposed model.
Some references to this would be welcome, it is proposed:
Ferrández-Pastor, F. J., Mora, H., Jimeno-Morenilla, A., & Volckaert, B. (2018). Deployment of IoT Edge and Fog Computing Technologies to Develop Smart Building Services. Sustainability, 10(11), 3832.
Apart from the previous comments, the article is well written and well-founded and promises to be a very useful document for the scientific community.
Minor comments:
The authors present a context-aware architecture in Figure 3. Please, explain if this architecture is developed originally by the authors or if it is based in other known architectures.
Figure 2. Fix the text in "Application Domains"
Author Response
Thank you for your comment.
We have included the reference you gave us and briefly compared our goals with those achieved using edge and fog computing.
We have restructured the paper. The architecture diagram is explained in section 4, we have explained in details how the different modules interact with the data stores in the architecture. We have included the thesis work of our co-author Zaki Chammaa in the references.
Reviewer 2 Report
This research is novel and very interesting, but will be improving with specific situations.
This research requieres more modern approach related with this topic as in:
Real-time video image processing through GPUs and CUDA and its future implementation in real problems in a Smart City AH Aguilar, JC Bonilla-Robles, JCZ DÃaz, A Ochoa International Journal of Combinatorial Optimization Problems and Informatics … 2019 |
And another this:
Yongrui Qin, Quan Z. Sheng:
Big Data Analysis and IoT. Encyclopedia of Big Data Technologies 2019
Hind Bangui, Mouzhi Ge, Barbora Buhnova:
Exploring Big Data Clustering Algorithms for Internet of Things Applications. IoTBDS 2018: 269-276
Is very important describes a comparative of this research with another similar -state of art- views.
Review the application domain to explain the future of this research.
Author Response
Thank you for your comments
The three references that you suggested are included in the list of references of the revised paper.
We have thoroughly revised the paper to improve the presentation of our research design, methodology, and the results obtained.
Reviewer 3 Report
The paper presents a model for data storage to be used in Smart City IoT implementations. The authors argue that a noSQL model, more precisely the MongoDB database is the most appropriate for such a setting. They also provide a model of representing IoT data using a MongoDB.
The paper is quite long and arguments these choices at length. However, the core of the paper is very simple. The authors prefer MongoDB over other databases. The use of noSQL databases is quite common and MongoDB is very popular in this type of setting; its advantages are well known. Arguing at length about it is superfluous.
What the authors should be focused on would be their choice of how to represent the data in the non-relational database. This is unfortunately lost in the mass of characteristics of MongoDB and/or non-relational databases.
There is no comparison between the proposed models and other models, in terms of the representation of data in the database.
The editing of the paper is not very good, some figures are squewed and the text is difficult to read, and the structure of the paper does not help the reader. A reorganization of sections and subsections is recommended.
The English of the paper is quite bad. It is full of grammar mistakes.
Some smaller comments:
The authors do not explicitly introduce BD as an acronym for Big Data.
In Figure 3 it is not clear why some arrows are not present and how does the data / control flow go from sensors to actuators. Some insights are given in the text, but a better visualization may be more appropriate. Who performs the data validation? Are these components or modules -- please make these concept names coherent throughout the paper.
"Since BD is heterogeneous, standard relational as relational data model and SQL DBMS, can be used". So, can a standard database be used or not?
Author Response
Thank you for your comments on our paper.
We have thoroughly revised our paper. It has been restructured, reduced, and refined.
We have made clear how the V-model characteristics of Big Data affects the different layer in IoT implementation, and motivated the necessity for suitable data stores for developing smart city applications. We have given rigorous justification to choose MongoDB, explained how things in IoT can be modeled in a generic manner, and then discussed things implementation in MongoDB. We have explained data integration process necessary for smart city application analysis. We have given the architecture that is currently being studied as the basis for smart city development in the thesis of our co-author Zaki Chammaa, and compared it with the current and ongoing related work.
We hope that all your concerns have been met satisfactory.
Round 2
Reviewer 3 Report
Many parts of the paper have been rewritten and clarified, which greatly increases the value of the paper.
However, there is still no actual comparison and no actual experimentation. When discussing the best database for extremely large smart city databases, at least a small scale experiment should be shown. Otherwise, a comparison only done on specifications is sterile.
The elements in Figure 13 are called Modules in the Figure, but Components in the text. This has not been corrected.
The text in Figure 3 is still unreadable. Similarly, Figure 12 is unreadable.
The English still needs to be revised, especially in the new text.
The phrase "tetra bytes" appears in the paper.
Author Response
All your comments have been addressed and the paper is revised.